# The Effect of Glacier Recession on Benthic and Pelagic Communities: Case Study in Herve Cove, Antarctica

**Marta Potocka** [1,*], **Anna Kidawa** [1], **Anna Panasiuk** [2], **Luiza Bielecka** [3],
**Justyna Wawrzynek-Borejko** [3], **Weronika Patuła** [3], **Kornelia A. Wójcik** [1],
**Joanna Plenzler** [1], **Tomasz Janecki** [4] **and Robert J. Bialik** [1]

[1]   Department of Antarctic Biology, Institute of Biochemistry and Biophysics, Polish Academy of Sciences,
      Pawinskiego St. 5a, 02-106 Warsaw, Poland
[2]   Department of Marine Plankton Research, Institute of Oceanography, Faculty of Oceanography and
      Geography, University of Gdańsk, Av. Marszalka Pilsudskiego 46, 81-378 Gdynia, Poland
[3]   Department of Marine Ecosystems Functioning, Institute of Oceanography, Faculty of Oceanography and
      Geography, University of Gdańsk, Av. Marszalka Pilsudskiego 46, 81-378 Gdynia, Poland
[4]   Nencki Institute of Experimental Biology, Polish Academy of Sciences, Pasteura St. 3, 02-093 Warsaw, Poland
*   Correspondence: mpotocka@ibb.waw.pl

**Abstract:** Changes in macrobenthic and pelagic communities in the postglacial, partially isolated, lagoon Herve Cove in Admiralty Bay, King George Island, were investigated 15 years after the first comprehensive studies had been conducted in this region. The bottom area of the cove has enlarged from approximately 12 ha to 19 ha after the retreat of the Dera Icefall. Based on a photographic survey of the benthos and taxonomic composition of zooplankton, ecological succession and the colonization of new species have been observed. Several new species occur such as gastropods, seastars, sea urchins and isopods, and their presence in different parts of the cove, as well as breeding aggregations suggests that they reproduce there. The influence of glacial streams is notable in bottom assemblages. We propose that Herve Cove is a good research area for studies on ecological succession in newly opened areas. The colonization of this lagoon has been recognized to be in its developing stage, and research should be continued.

**Keywords:** benthic fauna; biodiversity; colonization; environmental succession; King George Island; glacial retreat; macrozoobenthos; West Antarctica; zooplankton

## 1. Introduction

The Antarctic Peninsula region is one of the fastest-warming regions on Earth [1–3]. Recent results [4–9] have shown the rapid retreat of glaciers located on the South Shetland Islands. In the case of tidewater glaciers, this rapid retreat results in the uncovering of new, ice-free, unhabituated areas of the ocean floor [2,3,10,11]. Together, combined with retreating glaciers and the opening of ice-free areas, freshwater inflows and increases in mineral suspension in the water are the main consequences of global warming, and have an impact on organisms colonizing these habitats [12–14]. An increase in air temperature and, consequently, the faster melting of glaciers increases the processes affecting the supply of sediments to coastal waters, particularly as a result of progressive erosion [15]. In general, these sediments are released to coastal waters by melting icebergs or directly from a glacier terminus by subglacial or englacial meltwater outflows, which may further cause fluvial erosion of freshly exposed moraine on the forefields of these glaciers [16,17]. These factors have a significant impact on marine communities [12], especially on Antarctic bottom organisms, which are not well adapted to large amounts of suspended inorganic material [18].

On King George Island (the biggest island of South Shetlands archipelago), many small coves and proglacial lagoons have appeared due to the recession of glaciers; however, the lagoons are still under the direct influence of glaciers. The rate of change in these new ice-free areas is associated with fluctuations in air temperature [8,16]. Specifically, the last 40 or even 60 years of glacier recession in the South Shetlands Islands can be divided into four distinct periods: (1) from the end of the 1960s to the early 1980s, which showed a relatively stable warming trend; (2) approximately 20 years of a marked increase in the average air temperature with the warmest years in the late 1980s [19]; (3) a noticeable trend of cooling in 2000–2015 [8,20]; and (4) renewed warming since 2015–2016 [17].

Plankton and bottom communities from the central part of Admiralty Bay, the biggest bay on King George Island, were extensively studied, and showed great diversity in planktonic and benthic species [21–23]. At the same time, data from smaller, postglacial coves of this area (such as Goulden, Cardozo, and Herve coves) is scant [14,24]. Herve Cove was formed after the Dera Icefield retreated, when a large fragment of sea bottom appeared. The first comprehensive studies on the benthic communities in these area were conducted by Siciński et al. [25] in 1993. Significant differences between the composition of bottom and planktonic fauna of Herve Cove and the open waters of Admiralty Bay were noted, and attributed to the influence of tidewater glacier resting in the cove [25]. Since then, further retreat of Dera Icefall was observed. Therefore, Herve Cove offers an excellent opportunity to study the processes of colonization of the newly formed bottom area and water column volume. The main aim of the presented studies was to verify the hypothesis that newly opened areas formed by glacier retreats may be a location for the succession and colonization of benthic organisms and zooplankton; this verification was done using photographic surveys of benthic communities and the taxonomic composition of zooplankton. Relatively fast glacier retreats caused (1) the increased availability of new areas for colonization and (2) changes in the location of glacier streams that have impacts on environmental conditions. The results of our studies will provide additional data for the discussion about future trends in marine neritic ecosystems of polar regions.

## 2. Materials and Methods

### 2.1. Study Area

The research was conducted in 2008 *(benthic survey) and 2009 (plankton sampling)* in Herve Cove (Figure 1), which is located in the southern part of the Ezcurra Inlet, in the western branch of Admiralty Bay, King George Island (South Shetlands Islands, Antarctica). It is a small periglacial lagoon, typical for the shoreline of this area, and partially isolated from the Ezcurra Inlet by the submerged moraine [26]. The cove was created as a result of a gradual retreat of Dera Icefall [25].

In 2008, Dera Icefall was located high on the rocks, and did not have access to the sea. Two glacial streams were identified (Potocka, pers. obs. 2008)—one active on the western shore and one non-active (evidence of the old stream-bed) stream 100 m north from the active stream (Figure 1).

### 2.2. Glacier's Range

The front positions of the Dera Icefall have been determined using the following materials:

- Aerial photography taken on 20 December 1956 with a scale of 1:28,000 during the Falkland Island Dependency Aerial Survey Expedition (FIDASE);
- Aerial photography taken on 04 January 1979 with a scale of 1:3600 during the Second Polish Antarctic Expedition to H. Arctowski Polish Antarctic Station;
- Landsat 4 satellite images taken on 16 February 1990 with a resolution of 15 m.

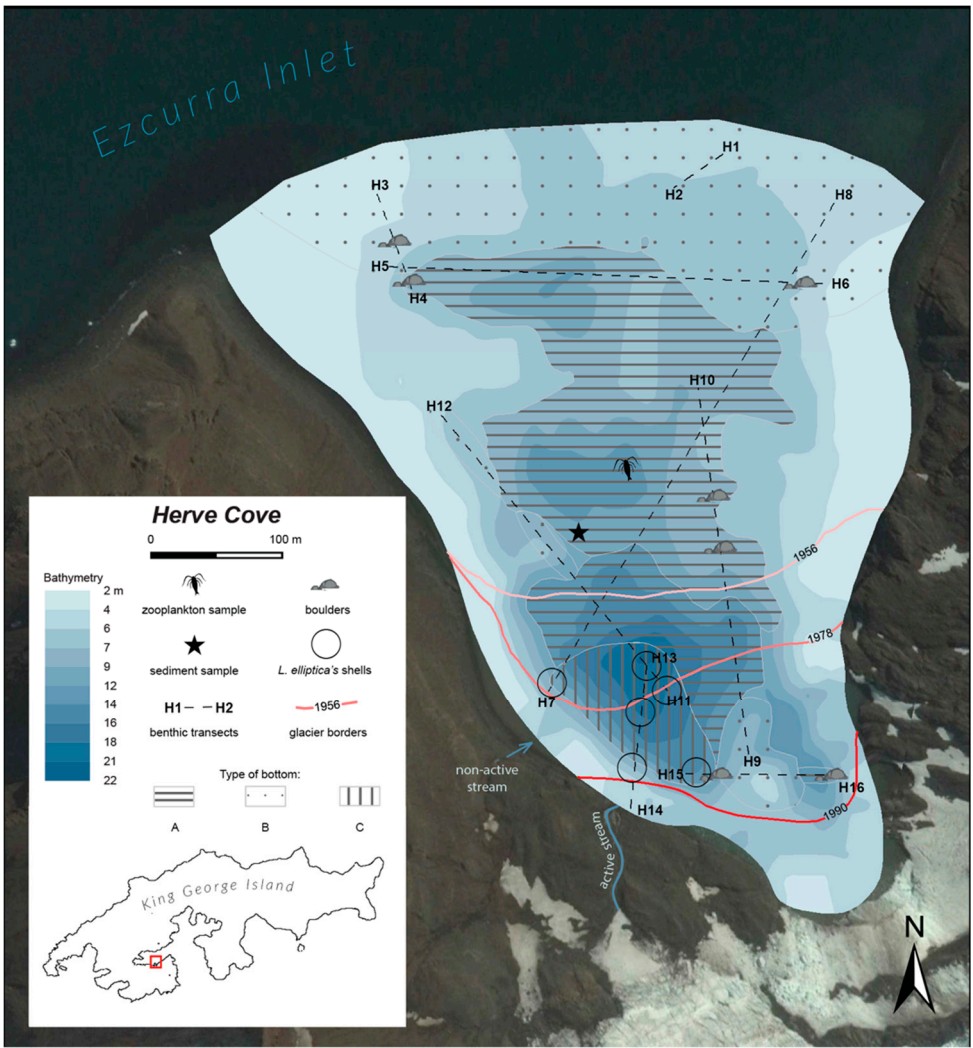

**Figure 1.** Study area, Herve Cove in Admiralty Bay, with glacier borders since 1956, the photo-transects along which the benthic communities surveys were made, and the plankton sampling station. Types of bottom recognized in the study area (A. soft bottom—sandy, silty and clay deposits, B. hard bottom—rocky and rocky-sandy deposits, C. soft bottom in the vicinity of streams), with boulders (B1) on soft bottom, bathymetry of the cove, and locations of empty shells of *Laternula elliptica* in the vicinity of glacier streams.

## 2.3. Bathymetry

Bathymetric measurements were done twice (4 January and 9 February 2018) with a Lowrance HDS-5 device (Lowrance Electronics, Tulsa, OK, USA), mounted on a RIB motorboat. The device was equipped with a vertical single-beam echo sounder (frequency 50/200 kHz, accuracy of 1% of the measured values, depth capability up to 1500 m). In addition, the dGPS provided the horizontal position with an accuracy of 3 m. The difference in the tidal level during both surveys was approximately 0.5 m (http://tides.mobilegeographics.com), which can be neglected because there were small waves during the measurement period and because the data were interpolated. Altogether, 4 km of profiles were performed, and based on these profiles, a bathymetry map was calculated using the spatial interpolation method provided within Dr Depth software 4.6. (Johnson Outdoors Marine Electronics, Inc., Eufaula, AL, USA) and ArcGis 10 (ESRI, Redland, CA, USA) software.

### 2.4. Sediments

To investigate the grain size distribution of sediment in Herve Cove, 1 kg of material was taken from the central part of the cove on 9 February 2018 with an Ekman bottom grab. The collected sediment was transported, drained, sieved, and weighed in the laboratory of the H. Arctowski Polish Antarctic Station. A mechanical shaker was employed for the gradation test with sieves from 0.5 mm to 32 mm.

### 2.5. Zoobenthos

Photographic survey by scuba-divers of benthos assemblages were carried out in January–February of 2008 along 8 linear photo-transects (Figure 1, Table 1). Sampling stations used by Siciński et al. [25] in 1993 were included for comparison of time-related changes in benthic communities.

**Table 1.** Photo-transect description: length, number of photo-stations, distance interval between photo-stations.

|  | Transects Length (m) | No. of Photo Stations | Intervals (m) |
| --- | --- | --- | --- |
| H1-H2 | 57 | 12 | 5 |
| H3-H4 | 55 | 8 | 7 |
| H5-H6 | 299 | 18 | 15 |
| H7-H8 | 400 | 42 | 10 |
| H9-H10 | 268 | 24 | 10 |
| H11-H12 | 261 | 35 | 7 |
| H13-H14 | 93 | 15 | 6 |
| H15-H16 | 121 | 17 | 7 |

A high-definition Olympus SP-550 UZ digital camera with Olympus PT-037 underwater housing and two LED lights was used. At each photo station, 3 photos (0.5–1 $m^2$ each) were taken at a distance of 80–100 cm from the bottom. No benthic individuals were collected or manipulated. The temperature and salinity at the surface and bottom of the photo-stations was measured during the photographic survey with a conductometer LF 197 (WTW GmbH). The distance from the glacier and streams was defined using GPS (Garmin GPSMAP 64st).

The macrobenthic animals and macroalgea visible on each photo were identified by two persons to reduce methodological bias. The image resolution was sufficient to identify macrobenthic organisms (>10 mm in diameter). Burrowing infauna were identified based on protruding parts. All recognized fauna was identified to the lowest possible taxonomic level, i.e., mostly species, and their abundance counted for each photo station. The determination of the exact area of the bottom surface captured in the photo was impossible due to the inability to use measurement frames as a result of the limited visibility caused by fine sediment. Only an estimation of the area was possible, and density of macrobenthic organisms is defined as a range of its abundance. Bottom types were characterized based on photos and personal observations and divided into rocky, sandy, and silty types. The estimated areas of each sediment type were determined based on the observations along transects with the photo stations and depth (Figure 1).

### 2.6. Zooplankton

Zooplankton samples were collected twice each month in January, February and March 2009 on one station in the central part of Herve Cove (Figure 1) with a WP-2 plankton net (mesh size 200 μm, inlet diameter 0.58 m), equipped with a flowmeter. Samples were preserved in 4% buffered formalin. During laboratory analysis, zooplankton was identified to the lowest possible taxonomic level and counted under a stereomicroscope (Nikon SMZ1500; NIKON SMZ 800, Nikon Instruments INC., Melville, NY, USA). Additionally, a group of unidentified chaetognaths was distinguished due to

the frequent damage to their bodies. Afterwards, the abundance of organisms was expressed as the number of individuals per 1000 m$^3$.

## 3. Results

### 3.1. Glacier Range, Bathymetry, Sediments

On a bathymetric map of Herve Cove (Figure 1), two clearly visible moraines were identified: One separating the cove from the Ezcurra Inlet (average depth 2–3 m), and one located in the central part of the cove, where the glacier front was situated in 1956 (depth of ca 6–8 m). Most of the bottom (82 photo stations occurring at depths 7–22 m) was covered by soft-bottom sediments (silty-clay-sand and sand), which was also confirmed by the sieve analysis of the sediment samples collected from the central part of the cove (coarse silt and fine sand, $d_{16}$ = 0.03 mm, $d_{50}$ = 0.09 mm and $d_{84}$ = 0.46 mm). Hard-bottom sediments (rocky and rocky-sand) were recognized at 24 photo stations (depths down to 7–9 m), mostly on the northern part of the cove, but also along the shallow eastern shore and part of the western shore, as well as on the moraine in the middle of the cove (Figure 1). On soft bottom, at depths of 6–10 m, 7 boulders were found (boulder with the adjoining area: between 0.5 and 1 m$^2$). The areas close to both streams (58 photo stations, depths down to 22 m) were covered by sand.

The water temperature ranged from 0.6 °C to 2.0 °C at the surface and from 0.8 °C to 1.7 °C at the bottom. The salinity ranged from 34.2 to 34.6 PSU at the surface (exception: area close to the active stream, where it ranged from 19.6 to 30.1 PSU with halocline noted at the depth of 0.2–0.5 m, up to 100 m from the stream), and from 34.3 to 34.6 PSU at the bottom (exception: area close to the active stream at 2 m depth, where it was 33.7 PSU).

### 3.2. Zoobenthos

A total of 13 benthic taxa (11 identified species) from seven phyla were observed in Herve Cove. Amphipoda (except *Cheirimedon femoratus* and *Bovallia gigantea*), Polychaeta (except *Spirorbis* sp.) and Bryozoa specimens were identified up to the family/species level (Table 2). Bottom fauna were dominated by mobile organisms, such as the amphipods *Ch. femoratus* and limpets *Nacella concinna*, and sessile organisms, such as sea anemones *Edwardsia* sp., bivalves *Aequiyoldia eightsii* (previously *Yoldia eightsii*) and *Laternula elliptica* (Figure 2). Individuals of *Ch. femoratus* were noted at 132 photo stations, while *A. eightsii* at 54, *Edwardsia* sp. at 48 and *L. elliptica* at 47. The number of species at each photo station ranged from 0 to 8.

Benthic organisms were assigned to three different groups based on the bottom sediments they were observed on: (A) soft-bottom (sandy, silty and clay deposits) assemblages, (B) hard-bottom (rocky and rocky-sandy deposits) assemblages, and (C) sandy bottom in the vicinity of streams (Figure 1, Table 2). The numbers of benthic species recognized on soft (A) and hard (B) bottom substrata were significantly different (Tukey's test HSD = 0.0414, $p < 0.05$), but the numbers of species on soft bottom (A) and soft bottom near the streams (C) did not differ significantly (Tukey's test HSD = 0.0621, $p > 0.05$).

The soft-bottom assemblage (A) (Figure 1; Figure 3A) consisted of seven taxa (Table 2), and the dominant organisms were sea anemones *Edwardsia* sp., amphipods *Ch. femoratus* and bivalves *A. eightsii* and *L. elliptica*. Another three species (*Odontaster validus*, *Parborlasia corrugatus*, *Sterechinus neumayeri*) were present sporadically. The number of species at each photo station ranged from 0 to 6 (mean = 3.1; SE = 0.13), In the central part of the cove (42 photo-stations), anemones *Edwardsia* sp. were scattered irregularly in numerous colonies (400–800 ind./photo station), together with amphipods *Ch. femoratus* (50–500 ind./photo station), but also with single bivalves. In other areas (36 photo stations), the dominant combination included the bivalves *A. eightsii* (50–200 ind./photo station) and *L. elliptica* (20–100 ind./photo station) occurring with *Ch. Femoratus*; anemones were present in small numbers ~5–10 ind./photo station.

**Table 2.** Benthic taxa present in assemblages recognized in Herve Cove. (A) Soft bottom—sandy, silty and clay deposits, (B) Hard bottom—rocky and rocky-sandy deposits, (B1) Boulders, (C) Soft bottom vicinity of stream.

| Phylum | Order | Genus/Species | Assemblage | | | |
|---|---|---|---|---|---|---|
| | | | A | B | B1 | C |
| Mollusca | Gastropoda | *Nacella concinna* | - | + | + | - |
| | Bivalvia | *Laternula elliptica* | + | + | - | + |
| | | *Aequiyoldia eightsii* | + | + | - | + |
| Echinodermata | Echinoidea | *Sterechinus neumayeri* | + | + | + | - |
| | Asteroidea | *Odontaster validus* | + | + | + | - |
| Cnidaria | Acinaria | *Edwarsia* sp. | + | + | + | + |
| Nemertea | Heteronemertea | *Parborlasia corrugata* | - | + | + | - |
| Arthropoda | Isopoda | *Glyptonotus antarcticus* | + | + | + | - |
| | | *Serolis polita* | - | + | - | - |
| | Amphipoda | *Cheirimedon femoratus* | + | + | + | + |
| | | *Bovallia gigantea* | - | - | + | - |
| Annelida | Polychaeta | *Spirorbis* sp. | - | + | + | - |
| Bryozoa | | Bryozoa uind. | - | + | + | - |
| | Macroalgae | | - | + | + | - |
| | Periphyton | | + | + | + | - |
| | Other *Notothenia coriiceps* | | + | - | - | - |

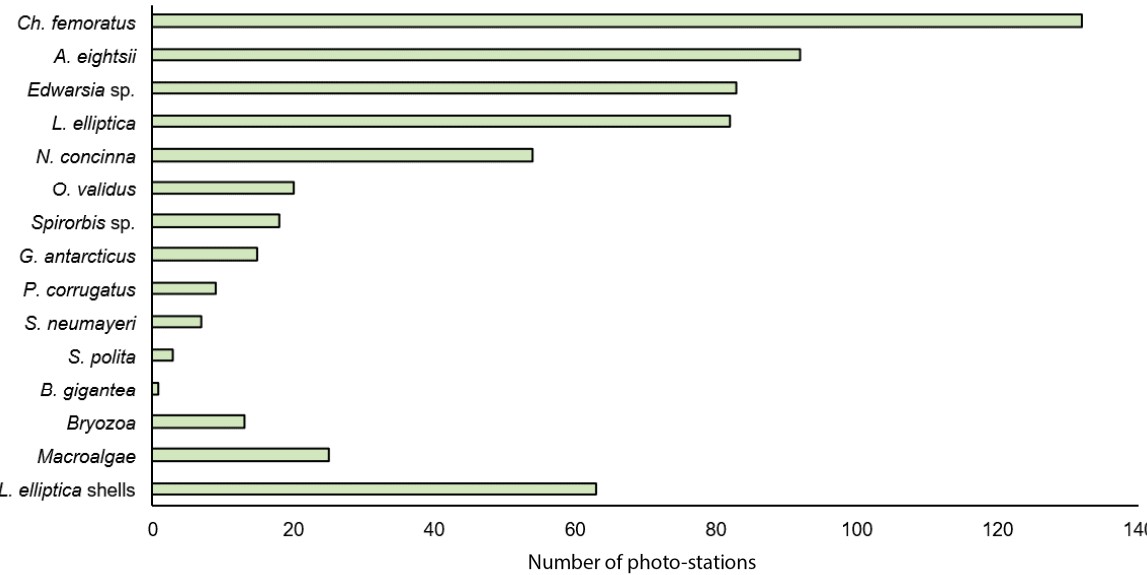

**Figure 2.** Benthic species occurrence at photo stations in Herve Cove.

Rocky and rocky-sandy bottom (assemblage B) (Figure 1; Figure 3B) was inhabited by 11 species (Table 2), and was the area with the highest species diversity. The number of species at each photo station ranged from 0 to 8 (mean = 4; SE = 0.4). Assemblage B was the most heterogenic, although the dominant combination (80% of photo stations) of *N. concinna*, *Spirorbis* sp. and Bryozoa was noted. Only limpets *N. concinna* (present at all photo stations) and polychaetes *Spirorbis* sp. occurred in higher

abundance (respectively 20–120 and 50–200 ind./photo-station). Macroalgae, such as *Ascoseira* sp., *Desmarestia* sp. and *Adenocystis* sp., covered 5–50% of that area.

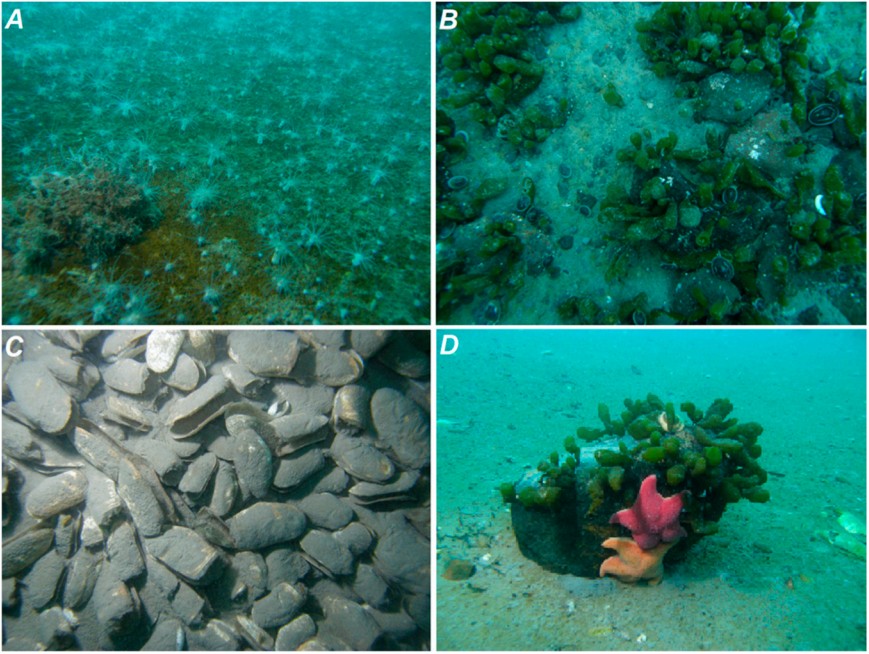

**Figure 3.** Benthic assemblages recognized in Herve Cove. (**A**) Soft bottom, (**B**) Hard bottom, (**C**) Soft bottom close to glacier stream, (**D**) Boulders (in text recognized as **B1**).

On the soft bottom (silty and clay), at depths of 6–10 m, assemblages concentrated around large boulders were recognized and assigned to a hard-bottom assemblage, as B1 (7 cases, Figure 1, Figure 3D). This group consisted of 10 species (Table 2), but most of them were present in small numbers (1–5 ind./photo-station). Limpets *N. concinna*, polychaetes *Spirorbis* sp. and amphipods *Ch. femoratus* were noted at each observed boulder in numbers of 1–31, 5–200 and 3–100 ind./boulder, respectively. The number of species at each photo station ranged from 4 to 7 (mean = 5.14; SE = 0.51). On the rocks or in their vicinity on the sediment, seastars *O. validus*, sea urchins *S. neumayeri*, nemerteans *P. corrugatus*, isopods *Glyptonotus antarcticus*, and limpets *N. concinna* were found and noted to be together with amphipods, such as *Ch. femoratus*, *B. gigantea* and other unidentified Polychaeta. Macroalgae (mainly *Ascoseira* sp., *Desmarestia* sp. and *Adenocystis* sp.) covered 30–80% of the analyzed boulders.

Assemblage C (sandy bottom) (Figure 1; Figure 3C), influenced by two streams (inactive and active), was located in the southwest part of the cove (depths from 7 to 22 m). This area had the lowest species diversity (only 4 species—Table 2). The number of species at a photo station ranged from 2 to 4 (mean = 2.6; SE = 0.1). High abundance of amphipods *Ch. femoratus* (from 20 to 200 ind./photo station) was noticed. A large number of empty shells (*L. elliptica*) was observed on the surface of the sediment in scattered groups of 10–200 shells (Figure 1; Figure 3C). Nevertheless, live bivalves *A. eighstii* (0–300 ind./photo station) and *L. elliptica* (0–90 ind/photo station) were also present. At photo stations with the highest numbers of empty *L. elliptica* shells, live *L. elliptica* were noticed sporadically, but *A. eightsii* were present in numbers ranging from 15 to 300 individuals. Anemones *Edwardsia* sp. were observed on 28 photo stations (1 to 30 ind./photo-station), and always occurred with amphipods *Ch. femoratus*.

In the center of the lagoon, two individuals of the fish *Notothenia coriiceps* were observed at depths of 16 m and 20 m. Several breeding aggregations of limpets *N. concinna* and two cases of mating *G. antarcticus* were noted in different areas of the lagoon. One anemone *Edwardsia* sp. feeding on salpa was also observed on the edge of the larger colony.

*3.3. Zooplankton*

A total of 26 taxa were observed in the investigated area (Table A1). Among the identified groups, representatives of Nematoda, Crustacea (Copepoda, Cumacea, Isopoda and Euphausiacea), Chaetognatha and Appendicularia were found. Furthermore, benthic larvae of Polychaeta, Gastropoda and Ascidiacea were also noted (Table 3). Among the copepods, the holoplankton structure consisted of 12 species (e.g., *Calanoides acutus*, *Metridia gerlachei*, *Oithona similis*) from three orders: Calanoida, Cyclopoida, and Harpacticoida (Table 3). Others were classified to a higher taxonomic level (Table A1). All found Harpacticoida were marked as unspecified. Regarding Euphausiacea, *Thysanoessa macrura* and *Euphausia crystallorophias* species were recorded. Cumacea was represented by one species (*Eudorella splendida*), and Appendicularia by one genus of *Fritillaria*. Additionally, three species of cheatognaths were found (*Eukrohnia hamata*, *Pseudosagitta gazellae* and *Solidosagitta marri*) (Table 3). Additionally, in January, we noted a clutch of pteropod eggs as well as a structure of an unidentified batch of eggs, presumably of benthic origin (not included in Table 3).

Representatives of Copepoda and Ascidiacea larvae were observed during the whole study period, while Nematoda, Gastropoda larvae, Isopoda, Cumacea, one species from Chaetognatha, *S. marri*, and one from Euphausiacea, *E. crystallorophias*, were noted only once throughout the entire sampling period (Table 3).

The highest abundance of planktonic organisms was noted at the end of January (810,000 ind.·1000 m$^{-3}$), of which 90% were identified as unspecified harpacticoids and cyclopoids (*O. similis*) (Table 3). The lowest abundance was observed two weeks later at the beginning of February (almost 36,000 ind.·1000 m$^{-3}$), with domination of calanoids and cyclopoids (Table 3); furthermore, two dominant species, *C. citer* and *O. similis*, had abundances greater than 24,000 ind.·1000 m$^{-3}$. Polychaetes and gastropod larvae were noted only in January but in small densities (502.7–1025.6 and 256.4 ind.·1000 m$^{-3}$, respectively) compared to ascidian larvae, whose abundances ranged from 210.8 to 4990.2 ind.·1000 m$^{-3}$ during the entire sampling period (Table 3). In January and March, the most abundant Copepoda were harpacticoids (in most cases, presumably benthic), whose contribution was above 60%. However, the contribution of cyclopoids was higher than that of others (>60%) on the other sampling days (Table 3). No clearly dominant species were observed in the euphausiids, and their numbers within the sampling days were very similar. On the other hand, in the chaetognaths, *E. hamata* was clearly dominant, both in number and in frequency.

**Table 3.** Main groups of zooplankton in Herve Cove. (A) Abundance (ind.·1000 m$^{-3}$), (%) Contribution.

| Phylum | Taxon | 13.01.2009 | | 22.01.2009 | | 05.02.2009 | | 20.02.2009 | | 05.03.2009 | |
|---|---|---|---|---|---|---|---|---|---|---|---|
| | | **A** | **%** | **A** | **%** | **A** | **%** | **A** | **%** | **A** | **%** |
| Nematoda | Nematoda | 0.0 | 0.0 | 0.0 | 0.0 | 0.0 | 0.0 | 475.3 | 0.4 | 0.0 | 0.0 |
| Annelida | Polychaeta larvae | 502.7 | 0.1 | 1025.6 | 0.1 | 0.0 | 0.0 | 0.0 | 0.0 | 0.0 | 0.0 |
| Mollusca | Gastropoda larvae | 0.0 | 0.0 | 256.4 | 0.0 | 0.0 | 0.0 | 0.0 | 0.0 | 0.0 | 0.0 |
| Arthropoda | Copepoda | | | | | | | | | | |
| | Calanoida | 15,932.8 | 3.7 | 38,974.4 | 4.8 | 8422.6 | 23.6 | 4039.7 | 3.7 | 14,545.1 | 3.8 |
| | Cyclopoida | 139,422.9 | 32.7 | 178,461.5 | 22.0 | 20,620.8 | 57.7 | 87,446.9 | 79.8 | 97,810.5 | 25.3 |
| | Harpacticoida | 268,738.1 | 63.1 | 583,333.3 | 72.0 | 1452.2 | 4.1 | 11,406.1 | 10.4 | 274,038.2 | 70.8 |
| | Isopoda | 0.0 | 0.0 | 0.0 | 0.0 | 290.4 | 0.8 | 0.0 | 0.0 | 0.0 | 0.0 |
| | Cumacea | 0.0 | 0.0 | 0.0 | 0.0 | 0.0 | 0.0 | 237.6 | 0.2 | 0.0 | 0.0 |
| | Euphausiacea | | | | | | | | | | |
| | *Thysanoessa macrura* | 128.1 | 0.0 | 256.4 | 0.0 | 0.0 | 0.0 | 0.0 | 0.0 | 210.8 | 0.1 |
| | *Euphausia crystallorophias* | 0.0 | 0.0 | 0.0 | 0.0 | 0.0 | 0.0 | 0.0 | 0.0 | 210.8 | 0.1 |
| Chaetognatha | *Eukrohnia hamata* | 0.0 | 0.0 | 2564.1 | 0.3 | 290.4 | 0.8 | 0.0 | 0.0 | 210.8 | 0.1 |
| | *Pseudosagitta gazelae* | 0.0 | 0.0 | 256.4 | 0.0 | 290.4 | 0.8 | 0.0 | 0.0 | 0.0 | 0.0 |
| | *Solidosagitta marri* | 0.0 | 0.0 | 256.4 | 0.0 | 0.0 | 0.0 | 0.0 | 0.0 | 0.0 | 0.0 |
| | Chaetognatha uind. | 630.8 | 0.1 | 1794.9 | 0.2 | 1161.7 | 3.3 | 0.0 | 0.0 | 0.0 | 0.0 |
| Chordata | Appendiculariae | 0.0 | 0.0 | 0.0 | 0.0 | 0.0 | 0.0 | 950.5 | 0.9 | 0.0 | 0.0 |
| | Ascidiacea larvae | 379.5 | 0.1 | 2820.5 | 0.3 | 3194.8 | 8.9 | 4990.2 | 4.6 | 210.8 | 0.1 |
| Total abundance (ind.·1000 m$^{-3}$) | | 425,734.9 | | 810,000.0 | | 35,723.3 | | 10,9546.3 | | 387,237.0 | |

## 4. Discussion

Herve Cove is the result of a gradual, continuous retreat of Dera Icefall, which was observed since 1956 [19]. The lagoon in front of the icefall has considerably grown in size up to the current size of 194,956 m$^2$ (total retreat from 1956 to 2008 amounts to 52,220 m$^2$). As a consequence of glacial retreat, a new extended environment was created for the benthic and planktonic communities. Survey of benthic and pelagic communities was done in 1993 [25], and gradual colonization of this area by benthic species was postulated [25].

To compare results of our studies with previous research [25] in Herve Cove area, methodological differences must be taken into account. Van Veen grab sampler used by Siciński et al. [25] samples fixed amounts of sediment, including infaunal organisms (area ~ 0.1 m$^2$ [2,22–24,27]. However, that method does not give a holistic picture of the spatial distribution of benthic organisms, mainly because we lose information on the spatial distribution of animals in a location that is being sampled, and this method cannot be used on rocky bottoms. Moreover, it does not allow to verify the presence of more mobile organisms, and causes partial devastation of the environment. Direct observation methods, such as the one used in presented studies, is more suitable to study epifauna, and, in particular, mobile epifauna, including all rocky bottom fauna [18,28,29]. However, it does not give information about i.e., infauna composition and biomass of benthic species. Due to method differences between our studies and the previous ones done in Herve Cove [25], only comparisons of macro-epifauna composition and their estimated abundance are discussed in this section. In the case of zooplankton studies, the methodology of collecting materials was very similar to that used by Siciński et al. [25] and Kittel et al. [30], with the exception of sampling time. In contrast to previous year-long researches, our samples were collected during the austral summer (January–March 2009).

All benthic and planktonic organisms found in Herve Cove are common species that have been noted on the same substrata and depths in other open areas of Admiralty Bay [14,30,31]. Although the number of planktonic taxa identified by us in Herve Cove was similar to that found 15 years ago [25], we have not found siphonophores, ostracods and pteropods (although pteropod egg clutches were present), and copepods were represented by a smaller number of species. Instead, we noted single individuals of euphausiids, cumaceans, isopods and appendicularians. These differences might be caused by the period of research (year-long research versus austral summer in our study). In comparison to data from 1993 [25,30], the maximum number of harpacticoids in our studies was almost 150 times higher. Similarly, in 2008, ascidian larvae occurred continuously from January to March and were much more numerous than they were 15 years before, when they were observed only in March [25]. The presence of unidentified batches of eggs, presumably of benthic origin, was noted, which might be explained by the high dynamics of waters of shallow coves, suspending some benthic organisms, such as harpacticoids, from different parts of Admiralty Bay.

Macrobenthic phyla identified in the present study represented only a small part of the taxa list recorded for Admiralty Bay—11 species in Herve Cove versus >150 species in Admiralty Bay [14,31], and significant benthic phyla such as Porifera, or classes i.e., Ophiuroidea or Ascidiacea present in Admiralty Bay are missing in Herve Cove [14,31]. Such phenomenon was observed in similar small glacial lagoons in Admiralty Bay [14]. Different observations were made in Marian Cove [4] and Potter Cove [28], two postglacial lagoons in Maxwell Bay (King George Island), where the most abundant groups were ascidians and bryozoans. In Herve Cove, the most abundant sessile macrobenthic taxa were anemones *Edwardsia* sp., whereas settled ascidians were not observed. The absence of adult Ascidiacea in Herve Cove is especially striking, as their planktonic larvae were observed in high numbers (almost 200 times more than in 1993, [25]). This taxa is known for their pioneering characters in newly opened areas [32]. The absence of adult ascidians may be explained by an insufficient amount of food (phytoplankton) [33] and/or by increased sediment runoff and high loads of particulate matter [3], which negatively affects filter-feeding organisms [34].

Herve Cove is smaller and shallower than both lagoons in Maxwell Bay. The inner part of Potter Cove in Maxwell Bay is less than 50 m depth, whereas the outer part is deeper than 100 m,

with a shallower sill separating them at a depth of <30m [28]. The morphological shape of cove might be an important factor influencing benthic colonization [35,36]. Transverse moraines are significant barriers [35] preventing homogenization of the communities from outside and inside the cove. The moraine in Herve Cove, at a depth of 2–3 m, separates the lagoon from the Ezcurra Inlet and can serve as a physical barrier for many organisms and their larvae, but also protects the bottom communities from destructive impact of icebergs. Ice cover in winter and glacial fresh water discharge in summer might affect inflow of pelagic larvae, decreasing biodiversity of the lagoon and increasing differences in species composition between different lagoons in the same area [35]. The second moraine in the middle of Herve Cove, occurring at depths of approximately 6–7 m, has probably less impact on the benthic distribution, as no differences between benthic communities on both sides of that moraine were observed.

In Herve Cove, three assemblages of benthic species were distinguished. Assemblages A (soft bottom) and C (influenced by the stream) were similar to those observed 15 years earlier by Siciński et al. [25]. No data on our third assemblage (B—hard-bottom) from 1993 is available, which is probably caused by methodological differences between the studies (Van Veen grab sampler versus photographic survey) and by a lack of sampling stations in these parts of the lagoon in 1993 [25]. The soft-bottom assemblage (A) was poorer in species than the hard-bottom assemblage (B). Similar observations were made in newly opened areas of Potter Cove, where rocky island uncovered by glacier retreat is characterized by high species richness [28].

Over 15 years, since the first studies of benthic fauna in Herve Cove [25], the expansion of several new species has been observed, together with an increase in species richness in the area. New species (sea urchin *S. neumayeri*, nemertini *P. corrugatus*, isopods *G. antarcticus* and *S. polita*) were noted. Dominance structure has not changed, but some of the dominant species expanded their range, e.g., *Edwardsia* sp. was noted in large colonies (>1000 ind./photo-station) in two additional areas south of the submerged moraine and in smaller colonies across the lagoon. These species are considered pioneer colonizers of newly opened or destroyed areas [37]. Two species (*N. concinna* and *O. validus*), which were previously only incidentally present near the submerged moraine [25], were now observed in comparatively large numbers (respectively >1200 ind. and 24 ind.) across the whole cove area of Herve Cove. In 1993, Siciński et al. [14] recorded the highest values of biomass of benthic species close to the submerged moraine (separating Herve Cove from the Ezcurra Inlet) and suggested the migration of these species from the outer side of the moraine into the lagoon. Our observations in 2008 showed changes in the benthic distribution in the hard-bottom area (assemblage B) in the cove. Species such as *N. concinna*, *O. validus*, *S. neumayeri* or *P. corrugatus* were observed inside the cove, even more than 400 m from the submerged moraine. However, colonization of the inside part of Herve Cove by migrating adult animals, except *Ch. femoratus* (mobile species, observed on all bottom types), seems to be impossible (bottom type diversity, distance of up to 400 m from the moraine). It seems unlikely that animals such as seastars, sea urchins, limpets or nemerteans could move from the Ezcurra Inlet to the far side of the cove, crossing areas of both soft and hard bottom. Breeding aggregations of limpets and mating *G. antarcticus* proved that at least some species breed inside Herve Cove. In the present study, we also found Gastropoda and Isopoda larvae, which were not observed in 1993 [25]. It is impossible to prove whether the larvae were derived from organisms that were already in Herve Cove or whether they migrated from Ezcurra Inlet. However, their presence suggests the further development of colonization of bottom fauna and more affordable conditions for several sensitive benthic species.

The presence of macroalgae was recorded at 15% of photo stations, and it was also a novelty for the area. Macroalgae in Antarctica provide shelter and food sources for large numbers of associated organisms [38]. This phenomenon was also observed in Herve Cove, where most mobile organisms present on hard bottom or around large boulders were congregated on macroalgae or in their vicinity. This may suggest the formation of new communities not previously observed in the cove and can play

an important role in shaping the diversity of faunal communities, as highlighted by Cacabelos et al. [39] and Torres et al. [40].

In Herve Cove, assemblages associated with the areas located close to the glacial streams (assemblages C) were, both in 2009 and 1993, species-poor, consisting of only four species: *A. eightsii*, *L. elliptica*, *Ch. femoratus*, and *Edwardsia* sp. In 2008, numerous empty shells and single live individuals of Bivalvia *L. elliptica* were observed there in scattered groups, down to the bottom at 22 m, whereas in 1993, no empty shells and high numbers of live specimens were noted by Siciński et al. [25]. *L. elliptica* is sensitive to meltwater streams and low salinity [41], which may explain their mass mortality in the area affected by the streams. Mass mortality of other bivalves, scallops *Adamussium colbecki*, was previously noted by Stockton [42] in McMurdo Sound. The author suggested that the death of scallops was due to the effect of the hyposaline lens from melting sea ice and freshwater runoff from the shore. The lower water salinity measured by us at a depth of 2 m near the active stream can serve as a proof of a similar scenario. The negative impact of freshwater inflow on glacial lagoon benthic ecosystems was also noticed by Stockton [42]; Barnes and Conlan [43]. However, the presence of empty shells at greater depths cannot be similarly explained, as the salinity there is almost constant (~34 PSU). In this case, another scenario should be considered, with large amounts of suspended mineral particulates carried with glacial meltwater being the main factor of the mass mortality. We assume the event occurred recently, as shells were still on the surface of the sediment, and previous studies [25] did not record their presence. In the same area (assemblage C), live *A. eightsii* were noted in large numbers, similar to 15 years earlier [25]. *A eightsii* feed on surface organic matter [44] and are more mobile than are *L. elliptica* [45], which make them better adapted to high sedimentation rates [46] common in glacial lagoons [27,47].

Based on the comparison of our studies with previous research done in 1993 [25], it can be assumed that the bottom communities of Herve Cove are progressing towards diversity typical for the open waters of Admiralty Bay. Continued comprehensive studies of the area are recommended to recognize further stages of succession of benthic and planktonic communities.

## 5. Conclusions

The Dera Icefall has retreated approximately 140 m, and a new bottom area of approximately 10,000 m$^2$ has appeared since the last comprehensive studies on benthic communities in Herve Cove occurred. New marine areas have been opened for colonization. An increase in bottom fauna biodiversity, relative to previous studies conducted 15 years ago, was also observed in this area. Several new species were found, including gastropods, sea stars, sea urchins or isopods, and their presence in different parts of the cove, as well as their breeding aggregations, suggest that they reproduce there. Most of the observed species have the ability to survive in conditions that are characteristic of glacier lagoons, such as high mineral suspension and non-stable salinity conditions. The succession of bottom communities depends on several factors, but one of the most important factors is the recruitment of the benthic species and factors influencing this phenomenon. Even without direct contact between the glacier and the water surface, its influence on bottom communities is still significant, especially in the area close to the meltwater discharge.

Probable scenarios of the colonization of new marine bottom for Herve Cove are as follows:

1.  Early colonization: Glacier retreated but still had contact with water; new bottom area opened but was still influenced by glacier discharge/meltwater/mineral suspension. Benthic animals can cross the submerged moraine and enter the lagoon, but conditions inside the lagoon are still unfavorable for settlement. Only tolerant species can survive in these conditions, such as anemones, amphipods and bivalves. This stage of colonization was most likely observed by Siciński et al. [25] in 1993.

2.  Colonization: Glacier retreated, and no contact with the water. The whole lagoon is open, and the influence of the glacier was less significant. Meltwater discharge and mineral suspension are significant seasonally and only affect small areas of the cove. Planktonic larvae of benthic species

can enter the cove and settle on the bottom. Macrofauna is also present at the far end of the cove, which provided favorable conditions for its larvae. There was higher biodiversity, but still a lack of many species. This stage of colonization was observed in 2008.

3. Late colonization/climax: Glacier has retreated far up the hill-side, no influence of meltwater. Colonization of more sensitive species. Biodiversity almost the same as that in the open water of the Ezcurra Inlet. The available climatic scenarios suggest that this bay should be an ideal place to analyze this stage of colonization in the near future, perhaps within 10–20 years.

**Author Contributions:** M.P., A.K., A.P. and T.J. designed the study. M.P., A.K., and T.J., provided benthic data. L.B., A.P., J.W.-B., and W.P. provided zooplanktonic data. M.P., A.K., A.P. conducted the majority of the data analysis. J.P., K.A.W., and R.J.B. made bathymetric, meteorological and glaciological observations and analysis. M.P. and A.K. wrote the first version of the paper. All authors contributed to the final version of the manuscript.

**Funding:** The work was financed by the Ministry of Scientific Research and Higher Education grant IPY/268/2006 (years 2007–2010) and by Department of Antarctic Biology Polish Academy of Sciences.

**Acknowledgments:** The authors would like to thank the crew of XXXII, XXXIII and XLII Polish Antarctic Expedition to the H. Arctowski Polish Polar Station, especially Michał Raczyński, Emil Nowicki, and Jerzy Borkowski, for their support during field studies. The authors would also like to thank Kazimierz Wielki University in Bydgoszcz for allowing us to use bathymetric probes. We would like to thank three anonymous reviewers for constructive advice that has improved our paper.

**Conflicts of Interest:** The authors declare no conflict of interest.

## Appendix A

**Table A1.** Comparison of recorded planktonic taxa in Herve Cove in 2009 (three-month study) to those in 1993–1994 (year-round studies). Gray color—differences between studies.

| Taxa | Our Station | Siciński et al., 1996 | Kittel et al., 2001 |
|---|---|---|---|
| Siphonophorae | - | + | + |
| Nematoda | + | + | + |
| Polychaeta | | | |
| *Pelagobia longicirrata* Greeff, 1879 | - | + | + |
| Polychaeta larvae | + | + | + |
| Gastropoda larvae | + | - | - |
| Gastropoda/Pteropoda | | | |
| *Limacina rangii f. antarctica* Woodward, 1854 | - | + | + |
| Ostracoda | | | |
| *Alacia belgicae* Müller, G.W. (1906) | - | + | + |
| *Alacia hettacra* Müller, G.W. (1906) | - | + | + |
| Copepoda | | | |
| *Calanus/Calanoides* (nauplii and copepodit I-III) | + | | |
| *Calanus propinqus* Brady, 1883 | + | + | + |
| *Calanoides acutus* Giesbrecht, 1902 | + | + | + |
| *Ctenocalanus citer* Heron et Bowman, 1971 | + | + | + |
| *Microcalanus pygmaeus* G.O. Sars, 1903 | + | + | + |
| *Euchaeta antarctica* Giesbrecht, 1902 | + | + | + |
| *Rhincalanus gigas* Brady, 1883 | - | + | + |
| *Racovitzanus antarcticus* Giesbrecht, 1902 | - | + | + |
| *Scaphocalanus* spp. | - | + | + |
| *Scolecithricella glacialis* Giesbrecht, 1902 | + | + | + |
| *Stephos longipes* Giesbrecht, 1902 | - | + | + |
| *Metridia gerlachei* Giesbrecht, 1902 | + | + | + |
| *Oithona frigida* Giesbrecht, 1902 | + | + | + |
| *Oithona similis* Claus, 1863 | + | + | + |
| *Oncaea antarctica* Heron, 1977 | + | + | + |
| *Oncaea curvata* Giesbrecht, 1902 | + | + | + |
| *Oncaea parila* Heron, 1977 | + | - | - |
| *Oncea* sp. | + | - | - |
| Harpacticoida | + | + | + |

**Table A1.** *Cont.*

| Taxa | Our Station | Siciński et al., 1996 | Kittel et al., 2001 |
|---|---|---|---|
| Isopoda | + | - | - |
| Cumacea | | | |
| *Eudorella splendida* Zimmer, 1902 | + | - | - |
| Euphausiacea | | | |
| *Thysanoessa macrura* G.O. Sars, 1883 | + | - | - |
| *T.macrura* – larvae | + | - | - |
| *Euphausia crystallorophias* Holt et Tattersall, 1906 | + | - | - |
| Chaetognatha | | | |
| *Eukrohnia hamata* Mobius, 1875 | + | + | + |
| *Pseudosagitta gazelae* Ritter-Záhony, 1909 | + | + | + |
| *Solidosagitta marri* David, 1956 | + | + | + |
| Chaetognatha uind. | + | | |
| Appendiculariae | | | |
| *Frittilaria* sp. | + | - | - |
| Ascidiacea larvae | + | + | - |

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
