# Peer review of "The Effect of Glacier Recession on Benthic and Pelagic Communities: Case Study in Herve Cove, Antarctica"

_jmse, doi:10.3390/jmse7090285_

Round 1
Reviewer 1 Report
Review of "The effect of glacier recession on benthic and pelagic communities: case study in Herve Cove, Antarctica."
This paper describes the pelagic and benthic community recruiting to newly exposed marine habitats after tidewater glacier retreat. The paper uses multiple methods to describe the area and community present in Herve Cove. The amount of effort involved is admirable, however the data are presented in an incoherent fashion, and no cohesive picture is created of the changing community, the effect of tidewater glaciers on benthic habitats, the effect of low salinity created by the now land based glacier runoff, and more.
Despite the importance of this type of work in remote, polar regions I would not accept this paper in its current presentation. I recommend rejection.
Major criticisms are as follows:
Abstract: no data is presented in this Abstract. This indicates that this is the first paper to study cleared space in Antarctica, which it is not, and lines 21-23 sound line part of the discussion, but deliver no information.
Introduction: This section is missing huge amounts of information:
- previous studies on tidewater glaciers in Arctic/Antarctic.
- there is no background to marine benthic ecology in Antarctica or around King George island.
- This ‘survey’ was previously done and there is no thorough description of that information.
- there is no background to the effect of freshwater input on marine systems- is this a major study objective?
- Colonization of marine benthos is not discussed at all
- Is r vs K selected species important or relevant in Antarctica?
- What about benthic habitat types- you survey many types, what would you expect benthos and communities to look like as a result of glacial retreat
Methods: These are confusing- multiple dates are used for time periods. None of these methods could be reproduced the way they are written.
- 1990-1993?
- 2007-2009
- 2018
- are there previous dates used?
- Is seasonality important in the surveys used? Unclear from descriptions
Figure1- this is results and methods… it’s a lot of info and cannot determine how you know the distribution of all sediment types.
SCUBA surveys are incomprehensive
Zooplankton collection methods and time periods are unclear.
How does your methodology mimic previous methods used in the area? How did you compare data sets? Were any statistics used to describe patterns in the benthic communities because they are not described.
One sample of sediment does not inform all sediment in the cove- not sure why this is included… or what the purpose was (for glacier runoff?).
Results: Salinity measurements are not described very well. What region, date, season, and depth are these taken at? A figure would be better.
You sample algae communities but never mentioned in the methods section. How important is this for benthic assemblages?
Overall the writing is hard to read and should be edited for clarity- largely because there is a large volume of species information and patterns are not clear.
Discussion-Conclusion: Like the Introduction there is a great deal of information missing- you could compare the results to those of other studies in Antarctica, Arctic, and other marine studies that look at colonisation and succession. There is not a lot of discussion about ecological processes.
Funding: It is not possible that no money was spent on this project. If this was opportunistically done with a different grant(s) that grant should still be reported along.
Author Response
Response to Reviewer 1 comments:
Comments and Suggestions for Authors
Review of "The effect of glacier recession on benthic and pelagic communities: case study in Herve Cove, Antarctica."
This paper describes the pelagic and benthic community recruiting to newly exposed marine habitats after tidewater glacier retreat. The paper uses multiple methods to describe the area and community present in Herve Cove. The amount of effort involved is admirable, however the data are presented in an incoherent fashion, and no cohesive picture is created of the changing community, the effect of tidewater glaciers on benthic habitats, the effect of low salinity created by the now land based glacier runoff, and more.
Despite the importance of this type of work in remote, polar regions I would not accept this paper in its current presentation. I recommend rejection.
Major criticisms are as follows:
Abstract: no data is presented in this Abstract. This indicates that this is the first paper to study cleared space in Antarctica, which it is not, and lines 21-23 sound line part of the discussion, but deliver no information.
We have added the most important data from our study to this section.
Introduction:This section is missing huge amounts of information:
- previous studies on tidewater glaciers in Arctic/Antarctic.
In the Introduction we mentioned about glacier recession in Antarctica, and South Shetlands Island. And, what is most important in our case - uncovering of new, ice-free, unhabituated areas of the ocean floor, which are discussed to be excellent area for colonization.
- there is no background to marine benthic ecology in Antarctica or around King George island.
We didn’t want to repeat information from different studies in the area. We cited several publications described benthic communities in Admiralty Bay and other areas in South Shetlands Islands i.e. Nonato, E.F., Brito, T.A.S., Paiva, P.C., Petti, M.A.V., Corbisier, T. N. Benthic megafauna of the nearshore zone of Martel Inlet (King George Island, South Shetland Islands, Antarctica): depth zonation and underwater observations. Polar Biol. 2000, 23(8), 580-588. Jażdzewski, K., De Broyer, C., Pudlarz, M., Zielinski, D. Seasonal fluctuations of vagile benthos in the uppermost sublittoral of a maritime Antarctic fjord. Polar Biol. 2001, 24, 910-917. Siciński, J., Jażdżewski, K., De Broyer, C., Presler, P., Ligowski, R., Nonato, E.F., Corbisier, T.N., Petti, M.A., Brito, T.A., Lavrado, H.P., BŁażewicz-Paszkowycz, M. Admiralty Bay Benthos Diversity—A census of a complex polar ecosystem. Deep Sea Res Part 2 Top Stud Oceanogr. 2010, 58(1-2), 30-48. Gaździcki, A., Majewski, W. Recent Foraminifera from Goulden Cove of King George Island, Antarctica. Pol. Polar Res. 2003, 24 (1), 3-12. Siciński, J., Rożycki, O., Kittel, W. Zoobenthos and zooplankton of Herve Cove, King George Island, South Shetland Islands, Antarctic. Pol Polar Res. 1996, 17, 221–238.
- This ‘survey’ was previously done and there is no thorough description of that information.
Previous studies of the Herve Cove area were done by Siciński et al (1996) and we do refer to these data several times, especially in the Discussion section. But as it was stated in Discussion, due to methodological differences between our studies and the previous ones done in Herve Cove (Siciński et al., 1996), we have focused on macro-epifauna composition.
- there is no background to the effect of freshwater input on marine systems- is this a major study objective?
We have added paragraph to the Introduction section. This is not a major study object. We have studied all aspects of impact of glacier recession on Herve Cove habitats.
- Colonization of marine benthos is not discussed at all
We do discuss it in the Discussion section.
- Is r vs K selected species important or relevant in Antarctica?
We have analyzed photos from each photo-station and we have noted species which are common and have been noted on the same substrata and depths in other open areas of Admiralty Bay.
- What about benthic habitat types- you survey many types, what would you expect benthos and communities to look like as a result of glacial retreat
We have added one paragraph to the Conclusion section – probable scenarios of further colonization of Herve Cove.
Methods:These are confusing- multiple dates are used for time periods. None of these methods could be reproduced the way they are written.
- 1990-1993?
- 2007-2009
- 2018
Dates of the research are described in the Methods section, and are as follow: Our research in Herve Cove was made in 2008 – benthic SCUBA diving survey, and in 2009 – plankton sampling. Bathymetric measurements and sediment sampling were done in 2018.
1993 it was the year when the first studies of that area were conducted by Siciński et al (1996), and we do refer to that results in our manuscript.
- are there previous dates used?
As above – we do refer to the previous studies in Herve Cove
- Is seasonality important in the surveys used? Unclear from descriptions
Distribution of benthic fauna in Antarctic waters does not depend on the seasonality, except for shallow zones exposed to ice. The influence of seasonality can be important only in case of planktonic species distribution, as it is discussed in the manuscript.
Figure1- this is results and methods… it’s a lot of info and cannot determine how you know the distribution of all sediment types.
We decided to use only one figure to show all the details, as we wanted the article to be concised. Sediment types were determined based on the SCUBA divers observations, and from photographic survey, and compare with previous research (Siciński et al 1996), and confirmed by the sieve analysis of the sediment samples collected from the central part of the cove.
SCUBA surveys are incomprehensive
Zooplankton collection methods and time periods are unclear.
In our opinion scuba diving survey was comprehensive, prepared in details and carried out as best as it was possible considering all aspects of diving in polar regions. Herve Cove is a small cove and the survey we have conducted there – 8 linear transects, 171 photo-stations located at regular intervals, were sufficient to describe the benthic composition of that area. To our transects we have also added the sampling stations from previous studies occurred in Herve Cove (Siciński et al. 1996) to compare benthic composition in the area.
Zooplankton sampling is described in Methodology section – samples were taken twice a month in January, February and March of 2009. The WP-2 plankton net (mesh size 200 µm, inlet diameter 0.58 m), equipped with a flowmeter, is a standard net uses in Admiralty Bay for zooplankton studies.
How does your methodology mimic previous methods used in the area? How did you compare data sets? Were any statistics used to describe patterns in the benthic communities because they are not described.
We have added the paragraph about differences in methodology of our studies (scuba diving survey) and previous research done by Siciński et al (1996) – van Veel grab. Due to methodological differences between our studies and the previous ones done in Herve Cove (Siciński et al., 1996), only comparisons of macro-epifauna composition and their estimated abundance are discussed in this section.In the case of zooplankton studies, the methodology of collecting materials was very similar to that used by Siciński et al. (1996) and Kittel et al. (2001), with exception of the sampling time. In contrast to previous year-long research, our samples were collected during the Austral summer (Jan-March 2009).
One sample of sediment does not inform all sediment in the cove- not sure why this is included… or what the purpose was (for glacier runoff?).
Sediment sample from the center part of the cove was taken to confirmed SCUBA divers observations of sediment distribution. As we stated in the Results section – most bottom were covered by soft-bottom sediment (silty-clay-sand), and it was confirmed by sieve analysis of the sample - coarse silt and fine sand, d16= 0.03 mm, d50 = 0.09 mm and d84= 0.46 mm. Hard-bottom sediments – rocky and rocky-sand, were described by SCUBA divers and confirmed on photos.
Results: Salinity measurements are not described very well. What region, date, season, and depth are these taken at? A figure would be better.
The temperature and salinity were measured at the surface and bottom of the photo-stations with a conductometer LF 197 (WTW GmbH), and it was during the SCUBA diving photographic survey. We have added that information to the manuscript (Methods).
You sample algae communities but never mentioned in the methods section. How important is this for benthic assemblages?
I have added that information to theMethodology section. We haverecorded the presence of macroalgae at 15% of photo-stations. Macroalgae provide shelter and food sources for large numbers of associated organisms – we do discuss that in Discussion section.
Overall the writing is hard to read and should be edited for clarity- largely because there is a large volume of species information and patterns are not clear.
We have edited some paragraphs and we believe that now the manuscript is more clear.
Discussion-Conclusion:Like the Introduction there is a great deal of information missing- you could compare the results to those of other studies in Antarctica, Arctic, and other marine studies that look at colonisation and succession. There is not a lot of discussion about ecological processes.
We have added some information to the Discussion section.
Funding:It is not possible that no money was spent on this project. If this was opportunistically done with a different grant(s) that grant should still be reported along.
We have added information about the project funds.

Reviewer 2 Report
Overall Comments: The manuscript is overall good, well written and clear. The data presented is new and the comparison between this new dataset and the survey conducted in 1993 in the same area is particularly interesting. I only have minor comments to provide.
Specific Comments:
Line 28 to 31: the sentence “Recent results … ocean floor” is too long and not very clear. I would suggest to the authors to separate it in two and reformulate it.
Line 31 to 33: glacier retreat and changes to water mineral concentration cannot be described as the main consequences of Global Changes on organisms. They are the consequences of Global Changes on polar physical environments which will in turn in pact the organisms colonising these environments. I would suggest the authors to rephrase their statement.
Line 42 to 43: during which time period was the plankton and bottom communities studied. Please specify.
Line 73: I am surprised to see that there are no recent records of the glacier to help determine the position of the Dezra Falls. The last record dates back from 1990. I am not an expert on this specific subject and so this might be quite ‘normal’ for these types of studies. How do other studies compare on this question?
Line 83: “mounted on a RIB motorboat. The device was…”
Line 85: “the dGPS provided…”
Line 93: It seems only on sediment sample was taken from a single location in the bay and used to characterise the sediment size distribution in the whole bay. Why were replicates not collected? Why only rely on a single sample?
Line 132: Could the location of the two moraines be added to Figure 1?
Line 152: This is the first time Ch. femoratus is mentioned in the text so please write the full genus name.
Line 183: Italicize Spirorbis sp.
Line 263: This might be obvious but I am struggling to understand the sentence “Presence of… from the bottom”. Why would resuspension of some benthic organisms explain the presence of unidentified eggs? Could the authors expand a bit on this ?
Line 279: Closing bracket “)” missing at the end of the sentence. I don’t know Maxwell Bay and I don’t find the information provided in the brackets clear. Could the authors maybe replace the bracket with a full sentence describing Maxwell Bay and how Herve Cove differs from it?
Line 288: Is the difference in the impact of the two moraines at Herve Cove only down to depth differences? Can the authors provide any other reason why this could be?
Author Response
Response to Reviewer 2 comments:
Comments and Suggestions for Authors
Overall Comments: The manuscript is overall good, well written and clear. The data presented is new and the comparison between this new dataset and the survey conducted in 1993 in the same area is particularly interesting. I only have minor comments to provide.
Specific Comments:
Line 28 to 31: the sentence “Recent results … ocean floor” is too long and not very clear. I would suggest to the authors to separate it in two and reformulate it.
We have added that information.
Line 31 to 33: glacier retreat and changes to water mineral concentration cannot be described as the main consequences of Global Changes on organisms. They are the consequences of Global Changes on polar physical environments which will in turn in pact the organisms colonising these environments. I would suggest the authors to rephrase their statement.
We have rephrased it.
Line 42 to 43: during which time period was the plankton and bottom communities studied. Please specify.
We have specified that information in the Ms. - benthic SCUBA diving survey, and in 2009 (Jan-March) – plankton sampling.
Line 73: I am surprised to see that there are no recent records of the glacier to help determine the position of the Dezra Falls. The last record dates back from 1990. I am not an expert on this specific subject and so this might be quite ‘normal’ for these types of studies. How do other studies compare on this question?
We have used the Landsat photography from 1990, as that was one of the last record of Dera glacier had contact with the water. The base photography of the Herve Cove is from 2014 and there is evidence that the Dera Icefall is high on the hill with no contact to the cove.
Line 83: “mounted on aRIB motorboat. The device was…”
Thank you, we have corrected that.
Line 85: “the dGPS provided…”
Thank you, we have corrected that.
Line 93: It seems only on sediment sample was taken from a single location in the bay and used to characterise the sediment size distribution in the whole bay. Why were replicates not collected? Why only rely on a single sample?
Sediment sample from the center part of the cove was taken to confirmed SCUBA divers observations of sediment distribution. As we stated in the Results section – most bottom were covered by soft-bottom sediment (silty-clay-sand), and it was confirmed by sieve analysis of the sample - coarse silt and fine sand, d16= 0.03 mm, d50 = 0.09 mm and d84= 0.46 mm. Hard-bottom sediments – rocky and rocky-sand, were described by SCUBA divers and confirmed from photos.
Line 132: Could the location of the two moraines be added to Figure 1?
We believe that the Figure 1 wouldn’t be clear to read after added two moraines. Both moraines are described in Results section, and can be detected on Fig1 via changes in bathymetry – first one separating the cove from the Ezcurra Inlet (the very North on Fig1), and one located in the central part of the cove where the glacier front was situated in 1956 (red line - glacier boarder from 1956).
Line 152: This is the first time Ch. femoratusis mentioned in the text so please write the full genus name.
That species was mentioned for the first time two lines above in brackets, and there was the full genus name.
Line 183: Italicize Spirorbis sp.
Thank you, we have corrected that.
Line 263: This might be obvious but I am struggling to understand the sentence “Presence of… from the bottom”. Why would resuspension of some benthic organisms explain the presence of unidentified eggs? Could the authors expand a bit on this ?
As the most of Antarctic benthic organisms have planktonic eggs and larvae, we believe that presence of egg batches in planktonic samples can be originally from other parts of the Admiralty Bay, not necessary from Herve Cove.
Line 279: Closing bracket “)” missing at the end of the sentence. I don’t know Maxwell Bay and I don’t find the information provided in the brackets clear. Could the authors maybe replace the bracket with a full sentence describing Maxwell Bay and how Herve Cove differs from it?
Thank you, we have corrected that.
Line 288: Is the difference in the impact of the two moraines at Herve Cove only down to depth differences? Can the authors provide any other reason why this could be?
First moraine (2-3m depth) is a physical barrier for organisms to explore Herve Cove, and the differences between benthic communities on both sides of the moraine are significant. Althought the second moraine (6-7m) in the center part of Herve Cove is not such a physical barrier – no differences in biodiversity or sediment on both sides of the moraine.

Reviewer 3 Report
The Ms bases on a comparative analysis of the benthic as well as the pelagic communities after the glacier recession. It analyzes the abiotic and biotic changes after 15 years in a smaller cove. The ms is well written, with basic but adaquate statistical analyses. The results are clear, and discussion is correct using up to date references.
Author Response
Response to Reviewer 3 comments:
Comments and Suggestions for Authors
The Ms bases on a comparative analysis of the benthic as well as the pelagic communities after the glacier recession. It analyzes the abiotic and biotic changes after 15 years in a smaller cove. The ms is well written, with basic but adaquate statistical analyses. The results are clear, and discussion is correct using up to date references.
Thank you for your review.
